# The association of presbyphagia with urinary incontinence and Incontinence-related quality of life in nursing home residents: A cross-sectional study

Ziya Yıldız 📧*

Department of Therapy and Rehabilitation, Uluborlu Vocational School, Isparta University of Applied Sciences, Isparta, Turkey

* ziyayildiz@isparta.edu.tr

## Abstract

### Purpose

The association between presbyphagia (age-related swallowing decline) and urinary incontinence (UI) remains poorly understood. This study therefore investigated the relationship between presbyphagia and both UI and incontinence-specific quality of life (I-QOL) in a distinct population of functionally independent healthy elderly in a nursing home.

### Method

The presence of presbyphagia was determined in 73 participants in the nursing home using the 100 ml water swallowing test (100mlSwal) and the Eating Assessment Tool-10 (EAT-10) tests. UI was determined using the International Consultation on Incontinence Questionnaire Short (ICIQ-SF) and I-QOL was determined using the King's Health Questionnaire (KHQ). The presence of presbyphagia, age and female gender confounding factors were analysed by multiple regression analysis. Multiple linear and binary logistic regression analyses were performed with presbyphagia, age, gender, and handgrip strength. Binary logistic regression was used to calculate odds ratios for UI presence.

### Results

Presbyphagia was significantly associated with UI (p = .001). Individuals with presbyphagia had 7.47-fold higher odds of UI. Presbyphagia was also significantly associated with multiple I-QOL sub-domains, including incontinence impact, role limitation, physical limitation, and emotions (all p < .001). Male gender was associated with higher UI presence (OR=5.16), but not with UI severity. Age showed no significant associations with UI or I-QOL.

**Data availability statement:** Data is shared via the Mendeley data system. https://data.mendeley.com/preview/s9z78b585n?a=3c-b7ca47-c4fc-46f3-bf99-d5098fc7f55d (The dataset is accessible via Mendeley Data. A free user account is required to log in and view the data, but registration is open to anyone and completely free of charge. There are no fees or any other restrictions for access. This complies with the PLOS ONE policy that requiring user registration is acceptable as long as anyone can register.).

**Funding:** The author(s) received no specific funding for this work.

**Competing interests:** The authors have declared that no competing interests exist.

## Conclusion

Presbyphagia is independently associated with UI presence and poorer I-QOL in nursing home residents. Male gender was associated with higher UI prevalence but similar severity and I-QOL to females. These findings suggest an independent association between presbyphagia and UI, thereby contributing to the formulation of hypotheses aimed at understanding the complex interactions between oral frailty and the urinary system. Further research is needed to elucidate underlying mechanisms and explore the potential clinical implications of this association.

## Trial registration

ClinicalTrials.gov NCT06827561

## 1. Introduction

Urinary incontinence (UI) and swallowing disorder (SD) are common problems in healthy elderly individuals living in nursing homes. One such SD is presbyphagia, defined as the structural, physiological, and innervational alterations in the swallowing process that occur as a result of healthy ageing. These age-related changes—including progressive brain atrophy, neurological deterioration, and loss of muscle mass and strength in the oropharyngeal region—can lead to clinical symptoms such as choking, coughing, pharyngeal residue, aspiration pneumonia and impaired swallowing rate [1,2]. Presbyphagia is an overlooked health problem in the elderly because it is perceived as part of the normal ageing process and develops asymptomatically [3]. It is the primary contributing factor to aspiration pneumonia in 50–75% of nursing home residents and is more common in this population compared to community-dwelling elderly [4].UI, which is considered a health priority by the World Health Organization, is another problem that emerges with ageing [5]. A multicentre study reported that the prevalence of UI in nursing homes was approximately 70% [6]. UI, which is caused by weakness in the nerve and muscle structure [7], is an important cause of poor quality of life in the elderly in nursing homes [5]. UI causes weaknesses such as decreased interpersonal relationships, loss of career and psychological well-being, limitations in the ability to exercise, and impaired sexual life [8].

The studies focusing on UI and SD in the elderly have been associated with sarcopenia through different mechanisms [8,9]. However, this relationship between UI and sarcopenia in elderly patients with SD was not found [10]. Theories such as the "soda pop model", which explains the anatomical relationship between the pelvic floor, diaphragm and pharynx/larynx, attempt to theorize the complex interaction between UI and swallowing function [11]. However, the literature explaining the relationship and mechanism between presbyphagia and UI and I-QOL remains limited. Although there is a study associating presbyphagia with lower urinary tract symptoms in middle-aged/elderly Japanese community-dwelling populations [12], research focusing on elderly individuals in nursing homes, particularly in middle-income countries like Turkey where institutional residency often reflects socioeconomic status rather

than health status, is notably absent. This population is particularly relevant given recent findings reporting a healthy ageing rate of 38.5% in Turkish nursing homes [13], indicating a substantial subgroup of healthy older adults in these settings.

In addition, effect modifiers and confounders associated with UI in nursing homes have not been adequately researched due to reasons such as facility characteristics, staffing issues, reimbursement and cultural attitudes [8]. Although UI is known to be hidden behind multiple diseases, weakness or other geriatric syndromes, [14] its relationship with presbyphagia has not been fully established. Therefore, this study aimed to investigate the relationship between presbyphagia and UI and I-QOL in this novel population of healthy elderly nursing home residents. Within this context, we tested the hypothesis that UI and I-QOL would be significantly associated with presbyphagia.

## 2. Method

### 2.1. Study design and setting

This cross-sectional observational study was conducted between December 23−30, 2024, using clinical tests and a face-to-face interview survey instrument. Participants provided written informed consent. The study included healthy elderly individuals aged 65−95 years at Gardenya Nursing Home and Hacı Ahmet Şençopur Nursing Home in Isparta, Turkey. The study was approved by the Research Ethics Committee of Isparta University of Applied Sciences on November 4, 2024 (CASE: E-96714346-050.04-139027, evaluation number: 203/5). The study was conducted in accordance with Declaration of Helsinki for research involving humans and the STROBE guidelines for conducting observational studies. The Mini mental state examination was used to determine the mental status of the participants. After demographic characteristics of the volunteers were recorded, 100mlSwal and EAT-10 scale were used to determine SD, the International Consultation on Incontinence Questionnaire Short Form (ICIQ-SF) was used to determine UI, and the King's Health Questionnaire (KHQ) was used to determine the I-QOL.

### 2.2. Participants and sample size

The required sample size was calculated a priori using G*Power 3.1 software. For a multiple linear regression analysis with a significance level (α) of 0.05, a statistical power (1-β) of 0.95, and a high effect size ($f^2 = 0.35$), a minimum of 40 participants was required. The study initially assessed 298 elderly individuals from the nursing home for eligibility.

The inclusion criteria were: (1) being a resident of the nursing home for at least one year, (2) a Mini-Mental State Examination (MMSE) score of 24 or above, indicating no significant cognitive impairment, (3) the absence of advanced hearing or visual problems that would hinder communication, and (4) provision of voluntary informed consent.

The exclusion criteria were designed to ensure a sample of functionally independent older adults without conditions that could directly confound the study's outcomes. Individuals were excluded if they met any of the following criteria: (1) dependency in any of the Basic Activities of Daily Living as assessed by the Katz ADL Index (index point<6), (2) a history of falls within the past 6 months or requiring assistance for ambulation, (3) a known history of neurological or psychiatric diseases (e.g., stroke, Parkinson's disease, dementia, or major depressive disorder defined by a score >10 on the 15-item Geriatric Depression Scale), (4) a history of diseases or surgeries with a known direct impact on swallowing or UI mechanisms, (5) regular use of medications that could significantly affect lower urinary tract function or cognitive status (e.g., diuretics, anticholinergics, antipsychotics), (6) an acute or uncontrolled systemic illness (e.g., uncontrolled hypertension, decompensated heart failure), (7) a diagnosed urinary tract infection within the past 4 weeks.

Based on these criteria, 225 individuals were excluded. The final study sample consisted of 73 participants. A post-hoc power analysis was conducted using the observed effect size ($f^2 = 0.17$) from the study data. This analysis confirmed that the study achieved a high statistical power of 97% at a 0.05 significance level, thus validating the robustness of the findings despite the observed effect size being lower than the initially anticipated high effect size used for the a priori calculation.

## 2.3. Measurements

EAT-10 scale and 100mlSwal test were used in the evaluation of presbyphagia. The 100mlSwal test was performed while participants were seated on a chair with armrests and asked to drink water quickly and not to hold it in the mouth. The time until the thyroid cartilage returned to its resting position was recorded for participants who had completely finished the water. The swallowing rate (ml/s) was calculated for all participants by dividing the amount of water drunk by the time recorded. Abnormal swallowing rate was considered to be below 10 ml/s. [15]. If the sum of the scores given to each item in the EAT-10 scale was 3 points or more, the risk of SD was accepted. Presbyphagia was considered to be present in healthy elderly individuals who scored 3 points or more on the EAT-10 scale [16] or who scored less than 10 ml/sec on the 100mlSwal test [15].

The ICIQ-SF is a scale that assesses the frequency, severity of UI. A Likert scale is used to assess the presence/absence and severity of symptoms. The overall score range is 0–21, with higher values indicating increased symptom severity. The cut-off score to distinguish between those with and without UI is 6 [17].

KHQ was used to calculate the change in I-QOL. In the first part, Role, Physical, and Social Limitations, Limitations in Personal Relationships, Emotional Problems, Sleep and Energy Disturbances are evaluated. The second part includes the Symptom Severity Scale, which assesses the presence and severity of urinary symptoms [18].

## 2.4. Statistical analysis

The data obtained were analysed using version 26.0 of the SPSS programme. Missing data were identified and corrected during data collection and entry. The skewness and kurtosis values of the data were checked for the assumption of normality, and the values were found to range between − 2.0 and + 2.0. The analysis was performed with presbyphagia/non-presbyphagia group, age, handgrip strength and gender as independent variables and UI and I-QOL as dependent variables. Multiple linear regression analysis was performed to explain the relationship between the variables. The R-squared ($R^2$) statistic evaluates the extent to which the regression model explains the variation in the dependent variable. The unstandardized beta coefficient (B) indicates the magnitude and direction of the relationship between the independent variables and the dependent variable. Pearson correlation and point-biserial correlation analysis were performed to determine the relationship between variables. Bonferroni correction was applied selectively to balance the risks of type 1 and type 2 errors. For the 10 KHQ subdimension—a coherent set of related outcomes assessing incontinence-specific quality of life—the significance threshold was set at $p < 0.005$ (0.05/10). The ICIQ-SF analysis for UI presence was considered a separate, independent hypothesis and was not subjected to multiple comparisons correction, consistent with standard practice for distinct research questions ($p < 0.05$). Multivariable Binary logistic regression was used to assess the association between presbyphagia (0 = non-presbyphagia, 1 = presbyphagia) and UI presence (ICIQ-SF ≥ 6; 0 = no, 1 = yes), adjusting for age and gender. Handgrip strength was excluded from the regression models due to high multicollinearity with gender ($r = −.959$; VIF = 12.55 and 12.54 for gender and handgrip strength, respectively), which exceeded the recommended threshold of 5 and could compromise model stability [19]. Results are presented as adjusted odds ratios (OR) with 95% confidence intervals (CI).

## 2.5. Validity and reliability

The questionnaires were administered by one researcher to ensure consistency. The internal consistency of the Turkish EAT-10 scale was found to be 0.90 Cronbach's alpha and the test repeatability was 0.91 Cronbach's alpha [16]. The ICIQ-SF is a scale with a validity level of A. The reliability and validity of the Turkish version were calculated as Cronbach's alpha value of 0.71 [17]. The internal consistency (Cronbach's alpha value ≥0.68) and test-retest reliability of the Turkish version of the KHQ were found to be high [18].

## 3. Result

A total of 73 white elderly, 35 males and 38 females, participated in the study. The study included 32 participants with presbyphagia (16 males, 16 females) and 41 participants without presbyphagia (19 males, 22 females). The mean age of the participants was $76.53 \pm 8.05$ years. The mean body mass index was $24.08 \pm 1.2$ kg/m$^2$. The mean swallowing rate was $15.4 \pm 5.32$ ml/sec. The total ICIQ-SF score was $5.68 \pm 4.13$ and Symptom Severity Scale was $6.45 \pm 5.92$. The 100mlSwa test rate was $15.4 \pm 5.32$ ml/sn (Table 1). There were 32 participants diagnosed with presbyphagia, including 4 participants with a 100mlSwa test rate below 10 ml/sec, 19 participants with an EAT-10 score of 3 and above, and 9 participants who failed both tests.

The UI status of the participants is shown in Fig 1. Based on the ICIQ-SF question 'When does urine leak?', 16.44% of participants (n = 12) reported no urine leakage. Among the 83.56% of participants (n = 61) who experienced some form of incontinence, the most frequently reported circumstances were: leakage when you are asleep (20.55%, n = 15), leakage when coughing or sneezing (17.81%, n = 13), leakage during physical activity (15.07%, n = 11), leakage when you have finishing urination and are dressed (9.59%, n = 7), leakage before reaching the toilet (8.22%, n = 6), leaks all the time (6.85%, n = 5), and leakage for no obvious reason (5.48%, n = 4).

Multiple linear regression analysis was conducted to examine the association between UI and the independent variables: presence/absence of presbyphagia, age, and gender. In the multiple linear regression analyses with UI and I-QOL as dependent variables, a high correlation was observed between gender and handgrip strength (r = −.959), and the VIF values were 12.55 and 12.54, respectively. Since these values exceeded the recommended threshold of 5, gender and handgrip strength could not be included simultaneously in any of the regression models [20].To ensure model reliability, handgrip strength was removed from the analyses.

The multiple linear regression analysis revealed that the model including presbyphagia, age, and gender was significantly associated with UI [$F_{(3,69)}$= 5.33, p = 0.002], explaining 15.3% of the variance. Accordingly, presence/absence of presbyphagia was significantly associated with UI ($\beta$ = 0.39, $t_{(69)}$=3.53, p = 0.001, pr$^2$ = 0.15), while age ($\beta$ = −0.02,

**Table 1. General information of the participants.**

|  | n | Minimum | Maximum | X±SS |
|---|---|---|---|---|
| Age (year) | 73 | 65 | 92 | 76.5±8.05 |
| ICIQTot | 73 | 0 | 18 | 5.68±4.13 |
| 100mlSwal (ml/sec) | 73 | 6.7 | 27 | 15.4±5.32 |
| EAT-10 | 73 | 0 | 7 | 2.31±1.61 |
| Handgrip strength | 73 | 15.7 | 32.1 | 23.14±5.72 |
| GeHea | 73 | 0 | 75 | 30.47±22.53 |
| InIm | 73 | 0 | 100 | 21±24.53 |
| RoLim | 73 | 0 | 66.67 | 8.44±13.35 |
| PhyLim | 73 | 0 | 83.33 | 16.43±19.54 |
| SoLim | 73 | 0 | 50 | 14.1±15.7 |
| Emo | 73 | 0 | 88.89 | 19.39±25.32 |
| SleEne | 73 | 0 | 83.33 | 31.5±27.71 |
| SevMea | 73 | 0 | 73.33 | 19.45±18.11 |
| SymSevSca | 73 | 0 | 21 | 6.45±5.92 |

X±SS: Mean± Standart Deviation, ICIQTot: ICIQ-SF total point, 100mlSwal: 100 ml water swalloving speed, EAT-10: Eating Assessment Tool-10 GeHea: General Health InIm:Incontinence impact RoLim: Role Limitation PhyLim:Physical limitation SoLim: Social limitation Emo: Emotions SleEne: Sleep/energy SevMea: Severity Measurements SymSevSca: Symptom Severity Scale

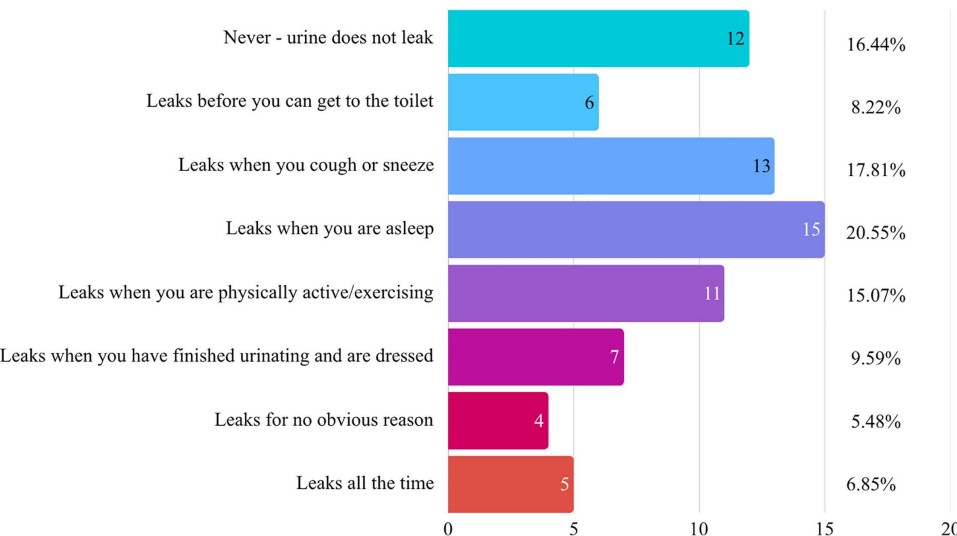

**Fig 1. Distribution of urinary incontinence patterns among nursing home residents (n = 73).** Values represent percentage of the total sample and number of participants.**.

t(69)=−0.18, p=0.85, pr²=0.0004) and gender (β=0.2, t(69)=1.85, p=0.06, pr²=0.04) showed no significant association (Table 2).

Multivariable binary logistic regression analysis revealed that participants with presbyphagia had 7.47 times higher odds of UI (OR = 7.47, 95% CI: 2.24–24.94, p=0.001) compared to those without presbyphagia, after adjusting for age and gender. Male gender was also significantly associated with higher UI odds (OR = 5.16, 95% CI: 1.53–17.42, p=0.008), while age was not significant (OR = 0.96, 95% CI: 0.90–1.04, p=0.329). The model was statistically significant (χ²=21.53, df=3, p<0.001) and correctly classified 71.2% of cases (Table 3).

Multiple linear regression analysis revealed that the model including presbyphagia, age, and gender was significantly associated with all I-QOL sub-dimension, (except for the sleep/energy) explaining between 9.6% and 20% of the variance. Accordingly, the presence/absence of presbyphagia was positively and significantly associated with several I-QOL sub-dimensions: general health (β=0.33, t(69) = 2.87, p=0.005, pr²=0.10), incontinence impact (β=0.41, t(69) = 3.71, p=0.004, pr²=0.16), role limitation (β=0.43, t(69) = 3.98, p=0.0001, pr²=0.18), physical limitation (β=0.39, t(69) = 3.62, p=0.001, pr²=0.16), social limitation (β=0.33, t(69) = 2.91, p=0.0048, pr²=0.10), emotion (β=0.41, t(69) = 3.70, p=0.0004, pr²=0.16), and symptom severity scale (β=0.34, t(69) = 2.67, p=0.004, pr²=0.09). No significant associations were

**Table 2. Regression coefficients of age, gender, presbyphagia group in ICIQ-SF.**

| Independent Variable | B [95% CI] | SE | β | p | Adjusted R |
|---|---|---|---|---|---|
| Constant | −.73 [−9.62, 8.16] | 4.45 | – | .87 | .15 |
| Presbyphagia | 3.28 [1.42, 5.13] | .92 | .39 | .001* | |
| Age | −.01 [−.12,.1] | .05 | −.02 | .85 | |
| Gender | 1.65 [−.12, 3.44] | .89 | .2 | .06 | |

R=0.43; R²=0.18; F=5.33 p=0.002 Durbin-Watson=1.66*, B: unstandardized regression coefficient β: standardized regression coefficient, SE standard error; (Reference categories: Presbyphagia=no; Gender=male)

**Table 3. Multivariabl Binary Logistic Regression Analysis for Factors Associated with UI Presence.**

| Variable | B (SE) | Wald | p | OR | 95% CI for OR | |
|---|---|---|---|---|---|---|
| | | | | | Lower | Upper |
| Presbyphagia | 2.01 (0.62) | 10.69 | .001 | 7.47 | 2.24 | 24.94 |
| Age | −0.04 (0.04) | 0.95 | .329 | 0.96 | 0.9 | 1.04 |
| Gender | 1.64 (0.62) | 6.97 | .008 | 5.16 | 1.53 | 17.42 |
| Constant | 1.59 (2.94) | 0.29 | .588 | 4.92 | | |

B = unstandardized coefficient. SE = standard error. OR = odds ratio. CI = confidence interval. ref: non-presbyphagia, (ref: female); Model fit: $\chi^2(3) = 21.53$, p < .001; Cox & Snell R² = .26; Nagelkerke R² = .35; correctly classified cases = 71.2%

found between age or gender and any KHQ sub-dimension (p > 0.005) (Table 4). There was no autocorrelation among the independent variables (1.5 < Durbin-Watson value > 2.5). There was no multicollinearity issue since handgrip strength was not included in the analyses based on VIF values.

The pearson correlation analysis revealed a negative and significant correlation between age and KHQ personal sub-dimensions (r = −.22 p = .05). Point biseral correlation analysis revealed a positive and significant relationship between Presbyphagia and ICIQ-SF total score (r = .38 p = .001) and KHQ sub-dimensions [general health (r = .3 p = .008), incontinence impact (r = .41 p < .001), role limitation (r = .44 p < .001), physical limitation (r = .41 p < .001), social limitation (r = .31 p = .006), emotions (r = .38 p = .001), severity measurements (r = .26 p = .02), symptom severity scale (r = .34 p = .003)]. In addition, a positive and significant correlation was found between gender and KHQ sub-dimensions [Social limitation (r = .23 p = .04), Emotions (r = .26 p = .02)] (Table 5).

## 4. Discussion

This study investigated the association between presbyphagia and UI and I-QOL in a distinct population of healthy elderly residents within a Turkish nursing home. The presence of presbyphagia was found to be independently associated with UI and poor I-QOL in healthy elderly people living in nursing homes. Notably, presbyphagia was independently associated with 7.47-fold higher odds of UI. Age was not found to have any association with UI and I-QOL. There was no significant association between the gender variable and I-QOL. In contrast, binary logistic regression analysis examining the categorical presence of UI revealed that male gender was independently associated with a 5.16-fold higher odds of UI compared to females. This study reveals an independent association between presbyphagia and UI, providing a basis for hypotheses on the interplay between oral frailty and urinary health.

Yoneyama et al reported a strong association of presbyphagia with UI independent of age (12). The results of our study are consistent with those of Yoneyama et al. This association may be related to anatomical and functional similarity of autonomic nerve fibers in the gastrointestinal and genitourinary systems, particularly the regulatory role of the autonomic nervous system on detrusor muscle, sphincter control [21] and oropharyngeal muscles [22]. Another factor that may contribute to the association between UI and presbyphagia may be inadequate fluid intake due to SD. Elderly people with presbyphagia may consume less fluid to avoid complications such as coughing or choking. This may have potentially impaired the optimal function of the bladder and contributed to mucosal dryness and increased urine density [23].

De Jesus Fidyk et al. reported a strong association with UI in nursing home residents who had difficulty in swallowing solid food using questionnaires [7]. Given the early stage uncertainty of presbyphagia and its tendency to be overlooked clinically [3]; we considered that their study may have included participants with severe swallowing dysfunction. In our study of individuals diagnosed with early stage presbyphagia but without significant symptoms, we observed a similar association to that foundby these authors. Thus, our findings suggest that the association between presbyphagia and UI might begin at an earlier stage than previously thought. They alsoreported that the odds of UI were 13.8 times higher in participants with dysphagia who stayed in a nursing home for two years or longer, while this rate was 7.47 times the odds

**Table 4. Regression coefficients of age, gender, presbyphagia group in KHQ sub-dimension.**

| Dependent Variable | Independent Variable | B [95% CI] | SE | β | p | AdjR | F |
|---|---|---|---|---|---|---|---|
| General health | Constant | 11.30 [−38.79, 61.39] | 25.11 | --- | .65 | .096 | 3.55 |
| | Presbyphagia | 15.01 [4.58, 25.45] | 5.23 | .33 | .005* | | |
| | Age | −.19 [−.84,.44] | .32 | −.07 | .54 | | |
| | Gender | 8.43 [−1.59, 18.45] | 5.02 | .18 | .09 | | |
| Incontinence impact | Constant | −27.41 [−76,52, 24.69] | 26.12 | --- | .29 | .17 | 6.1 |
| | Presbyphagia | 20.23 [9.37, 31.09] | 5.44 | .41 | .0004* | | |
| | Age | .05 [−.61,.72] | .33 | .01 | .87 | | |
| | Gender | 9.49[−.48, 20.38] | 5.22 | .2 | .06 | | |
| Role Limitation | Constant | −18.86 [−47.07, 9.33] | 14.13 | --- | .18 | .18 | 6.44 |
| | Presbyphagia | 11.74[5.86, 17.61] | 2.94 | .43 | .0001* | | |
| | Age | .06 [−.30,.42] | .18 | .03 | .74 | | |
| | Gender | 3.81[−1.83, 9.45] | 2.83 | .14 | .18 | | |
| Physical limitation | Constant | −35.59 [−76.68, 5.49] | 20.59 | --- | .08 | .19 | 6.69 |
| | Presbyphagia | 15.55 [6,99, 24.11] | 4.29 | .39 | .001* | | |
| | Age | .22[−.3,.75] | .26 | .09 | .4 | | |
| | Gender | 8.23 [.01, 16.46] | 4.12 | .21 | .05 | | |
| Social limitation | Constant | −8.32 [−42.69, 26.04] | 17.23 | --- | .63 | .12 | 4.43 |
| | Presbyphagia | 10.45 [3.29, 17.61] | 3.59 | .33 | .0048* | | |
| | Age | −.06 [−.50,.39] | .22 | −.03 | .80 | | |
| | Gender | 7.75 [.87, 14.63] | 3.45 | .25 | .03 | | |
| Personel Relationship | Constant | 12.08 [1.65, 22.51] | 5.23 | --- | .02 | .02 | 1.47 |
| | Presbyphagia | .68 [−1.50, 2.85] | 1.09 | .08 | .54 | | |
| | Age | −.14 [−.27,.003] | .07 | −.25 | .04 | | |
| | Gender | −.40 [−2.49, 1.69] | 1.05 | −.04 | .70 | | |
| Emotions | Constant | −21.43 [−74.60, 31.74] | 26.65 | --- | .42 | .19 | 6.76 |
| | Presbyphagia | 20.56 [9.48, 31.63] | 5.55 | .41 | .0004* | | |
| | Age | −.13 [−.81,.56] | .34 | −.04 | .72 | | |
| | Gender | 14.11[3.47, 24.75] | 5.34 | .28 | .01 | | |
| Sleep/energy | Constant | 36.42 [−28.56, 101.4] | 32.57 | --- | .27 | −.005 | .88 |
| | Presbyphagia | 10.28 [−3.25, 23.82] | 6.79 | .19 | .13 | | |
| | Age | −.33 [−1.17,.51] | .42 | −.09 | .44 | | |
| | Gender | 3.44 [−9.57, 16.45] | 6.52 | .06 | .6 | | |
| Severity Measurements | Constant | 17.94 [−22.25, 58.14] | 20.15 | --- | .38 | .1 | 3.65 |
| | Presbyphagia | 11.24 [2.86, 19.61] | 4.20 | .31 | .009 | | |
| | Age | −.35 [−.87,.17] | .26 | −.16 | .18 | | |
| | Gender | 7.95 [−.09, 16.00] | 4.03 | .22 | .053 | | |
| Symptom Severity Scale | Constant | −4.49 [−17.59, 8.60] | 6.56 | --- | .5 | .1 | 3.83 |
| | Presbyphagia | 4.02 [1.29, 6.74] | 1.37 | .34 | .0045* | | |
| | Age | .03 [−.14,.20] | .08 | .05 | .69 | | |
| | Gender | 1.71 [−.91, 4.33] | 1.31 | .15 | .20 | | |

B: unstandardized regression coefficient β: standardized regression coefficient, SE standard error; (Reference categories: Presbyphagia = no; Gender = male)

**Table 5. The correlation between dependent and independent variables.**

|  |  | Age[a] | Presbyphagia[b] | Gender[b] |
|---|---|---|---|---|
| ICIQTot | r | .08 | .38 | .18 |
|  | p | .46 | .001* | .11 |
| GeHea | r | .02 | .3 | .17 |
|  | p | .86 | .008* | .14 |
| InIm | r | .13 | .41 | .19 |
|  | p | .27 | .000* | .1 |
| RoLim | r | .15 | .44 | .12 |
|  | p | .19 | .000* | .27 |
| PhyLim | r | .2 | .41 | .2 |
|  | p | .08 | .000* | .08 |
| SoLim | r | .06 | .31 | .23 |
|  | p | .59 | .006* | .04⌐ |
| Personel | r | −.22 | .14 | −.05 |
|  | p | .05⌐ | .9 | .64 |
| Emo | r | .07 | .38 | .26 |
|  | p | .54 | .001* | .02⌐ |
| SleEne | r | −.04 | .15 | .05 |
|  | p | .7 | .17 | .65 |
| SevMea | r | −.06 | .26 | .2 |
|  | p | .56 | .02⌐ | .08 |
| SymSevSca | r | .13 | .34 | .13 |
|  | p | .24 | .003* | .25 |

r Pearson correlation, p test statistic value, [a] Pearson correlation analysis, [b] Point-biserial correlation analysis, *p < .001 ⌐p < 0.05

of UI in our study [7]. This difference may be related to the fact that long-term stay in a nursing home is associated with higher odds of UI [24]. The wide confidence interval observed in the multivariable binary logistic regression reflects the relatively small sample size and the exploratory nature of this study; however, the lower bound (2.24) remains substantially above 1, supporting the robustness and statistical significance of the association between presbyphagia and UI risk.

Previous studies have reported that swallowing and oral health disorders are associated with poor health-related quality of life (HrQOL). Oral and dental health problems related to presbyphagia have been linked to increased psychological burden, avoidance of social and physical activities, weakening of social relationships and a decreased overall life satisfaction, especially in the elderly in nursing homes. These studies have reported that impaired oral health triggers systemic diseases and dramatically reduces quality of life by disrupting nutrition [25,26]. Although there are studies in the literature investigating the association of SD with HrQOL, studies evaluating the I-QOL relationship are limited. In our study, the presence of presbyphagia was found to be associated with all sub-dimensions of I-QOL (except personal relationships, severity measurements and sleep sub-dimension). The participants' personal relationships may have been positively influenced by living with their peers in the nursing home and their sleep quality might be related to staying in single rooms.. The association between presbyphagia and I-QOL provides a hypothesis-generating foundation for future research investigating potential shared mechanisms underlying these conditions. Furthermore, given the association between presbyphagia and UI, we suggest that future studies investigating I-QOL consider the UI confounder.

UI is generally considered more common and associated with the female gender and older age, and most studies focus on this population [27,28]. Contrary to the traditional view, no relationship was found between age and UI in our study,

and the association with gender revealed a notable pattern. The study showed that gender had no significant association with the severity of urinary incontinence, but male gender was strongly and independently associated with the presence of urinary incontinence (OR = 5.16). This suggests that although UI prevalence is higher among male nursing home residents, its clinical severity is comparable between genders once incontinence develops. This finding aligns with previous studies reporting that gender differences in UI diminish with ageing in institutionalized populations [14,29]. Yavuz and Etiler reported that the male/female UI ratio was 1.4 in Turkish nursing home residents aged 65–74 years [29]. Similarly, Offermans et al. found no significant relationship between gender and UI in nursing home residents [8], and Aggazzotti et al. noted that UI risk increases in both genders at older ages [30]. A possible explanation for the higher UI prevalence in males may be urinary retention [31], but common risks shared by both genders—such as physical limitations [5], multiple drug use and pain [32] may have masked the gender association with UI severity.

The finding that male gender was associated with approximately 5-fold higher odds of UI in our study is consistent with Matsumoto et al. [33], who reported an independent association between chronic periodontal disease and lower urinary tract symptoms, particularly in males. Our study extends this finding from a relatively simple oral condition—periodontal disease—to a more complex oral dysfunction: presbyphagia. Both studies identify male sex as an important factor associated with UI, contrasting with the traditional focus on female gender in the general population. While they attributed this male-specific association to systemic inflammation-induced muscle weakness, our analysis—which included handgrip strength as a covariate— provides preliminary evidence that the presbyphagia-UI link is independent of generalized muscle weakness. These findings suggest that oral health in its broad spectrum-from periodontal status to swallowing function- may be relevant to UI, and that presbyphagia warrants greater clinical attention as a potential indicator of UI risk in especially in older male adults.

As with gender, age also did not emerge as a significant factor associated with UI severity in our study. Although Kessler et al. reported an association between age and UI risk in individuals over 75 years of age [34], no significant association between age and UI was found in the participants of our study, whose mean age was 76.5 years. A possible explanation for this finding may be that environmental factors such as regular toilet access and staff support in a nursing home setting may mitigate the adverse effects of advanced age by alleviating UI symptoms. Our findings indicate that the relationship between presbyphagia and UI and I-QOL should be evaluated in conjunction with other factors.

This study has several limitations. First, the cross-sectional design precludes the establishment of causal relationships. Second, the data were collected from non-profit nursing homes in a single city in Turkey, which may limit the generalisability of the findings. Third, self-reported data are susceptible to recall and response biases. Moreover, because both the exposure (presbyphagia) and the outcomes (UI, I-QOL) were measured using the same self-report, symptom-based instruments, shared method bias cannot be excluded. It is also possible that general symptom reporting tendencies—i.e., individuals who tend to report more symptoms in general—may have contributed to the observed associations.Fourth, medical records pertaining to swallowing complications were inconsistent. Fifthly, hydration status was not assessed in this study. Sixth, although we attempted to control for sarcopenia by including handgrip strength as a proxy measure, high multicollinearity between handgrip strength and gender (VIF > 5) necessitated its exclusion from the final regression models; therefore, residual confounding related to generalized muscle weakness as well as unmeasured factors such as frailty and overall functional decline, cannot be completely ruled out and may have influenced the observed associations..Seventh, the inability to interview all residents with cognitive impairments represents an additional limitation. Lastly although data on UI subtypes were collected, the small number of participantsprecluded subtype-specific statistical analyses, and the inability to differentiate between stress and urgency UI subtypes—each possibly involving distinct pathophysiological mechanisms—further limits interpretation.

## 5. Conclusions

This study provides novel evidence that presbyphagia is independently associated with UI and poorer I-QOL among functionally independent elderly nursing home residents. Individuals with presbyphagia had 7.47-fold higher odds of UI

compared to those without presbyphagia. The association, which persisted after accounting for age, handgrip strength and gender, challenges the conventional focus on these traditional risk factors in institutional settings and suggests that oral frailty may offer a different perspective on pelvic health in the elderly. Regarding gender-related findings, while male gender was associated with approximately 5.16 -fold higher odds of UI presence, no significant gender differences were observed in UI severity, nor was gender significantly associated with I-QOL. This suggests that once incontinence develops, its clinical course is similar in both genders. Additionally, age was not significantly associated with UI presence, UI severity, or I-QOL.

However, several limitations should be acknowledged, including the modest sample size, single-centre design, inability to perform UI subtype-specific analyses due to low urgency UI prevalence, lack of hydration status measurement, and exclusion of handgrip strength due to multicollinearity with gender. Additionally, both presbyphagia and UI outcomes were assessed using symptom-based questionnaires, which introduces the possibility that the observed associations may partially reflect overall symptom burden or response bias rather than a specific physiological link. Furthermore, residual confounding from unmeasured factors such as sarcopenia, frailty, or overall functional decline cannot be fully ruled out and should be considered when interpreting these findings. Despite these limitations, it should not be overlooked that our study is exploratory and hypothesis-generating in nature. Accordingly, our findings highlight the need for future research with larger, more diverse cohorts incorporating comprehensive assessments of potential confounders including hydration status, sarcopenia measures, and UI subtypes to better understand the relationship between presbyphagia and UI. This study contributes to the understanding of the complex interaction between presbyphagia and UI and provides a hypothesis-generating foundation for future research in this area.

## Acknowledgments

We would like to thank Ayşegül Yıldız for her proofreading of the manuscript and for facilitating institutional communication.

## Author contributions

**Conceptualization:** Ziya Yıldız.

**Data curation:** Ziya Yıldız.

**Formal analysis:** Ziya Yıldız.

**Funding acquisition:** Ziya Yıldız.

**Investigation:** Ziya Yıldız.

**Methodology:** Ziya Yıldız.

**Project administration:** Ziya Yıldız.

**Resources:** Ziya Yıldız.

**Software:** Ziya Yıldız.

**Supervision:** Ziya Yıldız.

**Validation:** Ziya Yıldız.

**Visualization:** Ziya Yıldız.

**Writing – original draft:** Ziya Yıldız.

**Writing – review & editing:** Ziya Yıldız.

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
