## [Decision Letter · Decision Letter 0]

2 Feb 2026

PONE-D-25-62171The Association of Presbyphagia with Urinary Incontinence and Incontinence-Related Quality of Life in Nursing Home Residents: A Cross-Sectional StudyPLOS One

Dear Dr Ziya Yildiz,

Thank you for submitting your manuscript to PLOS ONE. After careful consideration, we feel that it has merit but does not fully meet PLOS ONE’s publication criteria as it currently stands. Therefore, we invite you to submit a revised version of the manuscript that addresses the points raised during the review process.

We look forward to receiving your revised manuscript.

Kind regards,

Stanisław Jacek Wroński, M.D., Ph.D, FEBU

Academic Editor

PLOS One

Journal Requirements:

2. In the online submission form, you indicated that [Insert text from online submission form here].

3. We note you have included a table to which you do not refer in the text of your manuscript. Please ensure that you refer to Table 3 in your text; if accepted, production will need this reference to link the reader to the Table.

4. Please include a copy of Table 5 which you refer to in your text on page 9.

Reviewers' comments:

Reviewer's Responses to Questions

**Comments to the Author**

1. Is the manuscript technically sound, and do the data support the conclusions?

Reviewer #1: Partly

Reviewer #2: Yes

Reviewer #3: Yes

2. Has the statistical analysis been performed appropriately and rigorously? 

Reviewer #1: Yes

Reviewer #2: Yes

Reviewer #3: Yes

3. Have the authors made all data underlying the findings in their manuscript fully available?

Reviewer #1: Yes

Reviewer #2: No

Reviewer #3: No

4. Is the manuscript presented in an intelligible fashion and written in standard English?

Reviewer #1: Yes

Reviewer #2: Yes

Reviewer #3: Yes

5. Review Comments to the Author

Reviewer #1: First of all, I appreciate the opportunity to read your work. Studies that seek to identify intervening factors in health outcomes relevant to institutionalized older adults certainly contribute to improving the clinical practice of the professionals involved.

However, there is a central point that deserves greater attention: presbyphagia and urinary incontinence frequently coexist because both emerge from the same pathophysiological context. This shared set of determinants creates a scenario in which any observed association between the two conditions is likely explained by confounding factors, especially sarcopenia. Since this important confounder was not analyzed, even statistical adjustments cannot fully address this limitation, which compromises the interpretation of the results.

Therefore, I recommend restructuring the text so that the findings related to the dependent variables—presbyphagia and urinary incontinence—are presented in relation to quality‑of‑life measures independently, avoiding causal inferences between presbyphagia and incontinence.

Reviewer #2: This manuscript addresses a novel and underexplored association between presbyphagia and urinary incontinence (UI) among functionally independent nursing home residents. The topic is relevant, interdisciplinary, and appropriate . The study is generally well written, ethically sound, and methodologically transparent. However, several methodological, analytical, and reporting issues need to be addressed before the manuscript can be considered for publication.

The abstract is slightly long and statistically dense. Abstract could be made little simpler

This is a cross-sectional study, but at several places the wording sounds like presbyphagia leads to urinary incontinence. Please be careful with language-Say “associated with”, not “predicts”, “affects”, or “should be screened as standard practice”. The conclusions and clinical suggestions should be more cautious and framed as hypothesis-generating, not practice-changing.

Definition of presbyphagia needs clearer justification

statistical analysis:The logistic regression model used to calculate the odds ratio is not clearly described

Confidence intervals for the odds ratio are missing

Figure 1 needs a clearer explanation in the text

Tables are very dense and heavy on abbreviations

Clearly state which analysis was used for the odds ratio.Report adjusted OR with 95% CI. Be consistent in how UI is analysed and interpreted

UI and swallowing function can also be influenced by:BMI, Mobility, Comorbidities, Hydration status, Medication burden.If these variables are unavailable, please explicitly acknowledge residual confounding as a limitation.

If some data exist (e.g., BMI), consider including them in an exploratory model.

Reviewer #3: Thank you for the invitation to review the manuscript entitled “The Association of Presbyphagia with Urinary Incontinence and Incontinence-Related Quality of Life in Nursing Home Residents: A Cross-Sectional Study.”

This is an original study that raises an interesting but conceptually challenging question. The primary aim is to demonstrate an association between presbyphagia and urinary incontinence. While such an association is statistically demonstrated, causation is not established, and the clinical or practical implications of this finding remain unclear. The results are statistically significant but conceptually weak in terms of mechanistic or clinical interpretation.

Several methodological concerns merit attention. Urinary incontinence is not a single clinical entity; stress and urgency incontinence have distinct mechanisms and etiologies. In the present study, incontinence appears to be treated as a unified condition and linked to presbyphagia as a single construct, which may oversimplify the clinical reality. In addition, the sample size is small and drawn from a single population, limiting generalisability.

Overall, the study is best interpreted as exploratory and association-based, rather than as a hypothesis-confirming or clinically directive investigation.

The main strengths of the manuscript include the use of validated questionnaires, clear reporting of regression coefficients and correlations, and the absence of obvious internal data inconsistencies. However, it would be more appropriate to replace the term “predicts” with “is associated with” throughout the manuscript when describing regression results.

Although I am not a statistician, the manuscript would benefit from formal statistical review. In particular, there appears to be a multiple comparisons issue, with approximately 30 hypothesis tests conducted without correction. Additionally, a logistic regression analysis is implied by the reporting of an odds ratio (OR = 4.61), but the corresponding model and confidence intervals are not presented.

Major Concerns

Conceptual overlap: Presbyphagia is defined using EAT-10, a symptom-based instrument, while outcomes include multiple symptom-based and quality-of-life scales (ICIQ-SF, KHQ). The observed associations may therefore reflect overall symptom burden, frailty, or response bias rather than a specific link between swallowing function and urinary continence. This limitation should be explicitly acknowledged, and conclusions reframed accordingly.

Causality: At several points, the manuscript implies a directional or causal relationship. These statements should be revised to consistently emphasise association only.

Clinical relevance: The variance explained by the regression models is modest (R² approximately 0.10–0.20). Statements suggesting clinical screening, diagnostic implications, or practice change should therefore be avoided.

Minor Concerns

If available, comorbidity or frailty measures should be considered as potential confounders.

Recruitment setting and exclusion criteria should be clarified.

In conclusion, the study demonstrates a statistical association between presbyphagia and urinary incontinence. Older adults classified as having presbyphagia score worse on ICIQ-SF, KHQ, and quality-of-life questionnaires. These findings are compatible with broader constructs such as frailty, comorbidity, and symptom burden.

However, the study does not establish a causal relationship, does not identify a shared physiological mechanism, and does not justify clinical screening for urinary incontinence based on presbyphagia alone.

The manuscript may be considered for publication provided the above issues are adequately addressed and the exploratory nature of the findings is clearly acknowledged.

6. PLOS authors have the option to publish the peer review history of their article (what does this mean?). If published, this will include your full peer review and any attached files.

Reviewer #1: **Yes:**Denise Pinheiro Marques Alves dos Santos

Reviewer #2: **Yes:**Amisha S Amin

Reviewer #3: No

---

## [Author Response · Author response to Decision Letter 1]

12 Mar 2026

Response to Reviewer #1

Reviewer #1: “…However, there is a central point that deserves greater attention: presbyphagia and urinary incontinence frequently coexist because both emerge from the same pathophysiological context. This shared set of determinants creates a scenario in which any observed association between the two conditions is likely explained by confounding factors, especially sarcopenia. Since this important confounder was not analysed, even statistical adjustments cannot fully address this limitation, which compromises the interpretation of the results. Therefore, I recommend restructuring the text so that the findings related to the dependent variables—presbyphagia and urinary incontinence—are presented in relation to quality‑of‑life measures independently, avoiding causal inferences between presbyphagia and incontinence.”

Answer: Thank you for your insightful comment regarding the shared pathophysiological background of presbyphagia and urinary incontinence, and the potential confounding role of sarcopenia. You have raised a very important point that we had not initially considered. In response to your valuable feedback, we recognized the need to control for sarcopenia in our analysis. Since handgrip strength measurements are routinely collected in our institution every three months and recorded in the institutional database, we obtained the values corresponding to the study period and attempted to incorporate them into our regression model as a proxy measure for sarcopenia. However, during the diagnostic process for multicollinearity, we encountered a critical statistical issue: a very high correlation between handgrip strength and gender (r = -.959) (Figure 1), resulting in Variance Inflation Factor (VIF) values of 12.55 and 12.54 (Figure 2), respectively. Since VIF values significantly exceeded the recommended threshold of 5 (Craney and Surles 2002), including both variables would have compromised the reliability and validity of our model. Consequently, despite our attempt to address your concern by obtaining and testing handgrip strength data, we had to exclude it from the final analysis due to these statistical limitations.

As noted by O'Brien (2007), when VIF values exceed recommended thresholds, researchers commonly address multicollinearity by removing one of the correlated variables, using ridge regression, or combining variables into a single index. Similarly, Kim (2019) emphasizes that excluding multicollinear predictors is an appropriate approach to obtain stable and reliable regression models. Following these established methodological recommendations, we chose to exclude handgrip strength from our analyses, as its high correlation with gender (r = -.959; VIF > 5) made simultaneous inclusion of both variables statistically problematic. This decision ensured the consistency and reliability of our regression models. As you recommended, we have now explicitly stated in the Limitations section that, although we attempted to include handgrip strength, sarcopenia could not be adequately controlled for as a potential confounder. Due to these changes and to avoid causal inferences, we have restructured the text.

Figure 1. Correlations (handgrip vs gender) (The Figure file format is not supported, so it could not be added here. The Figure can be viewed in the relevant Word document.)

Figure 2. VIF value (The Figure file format is not supported, so it could not be added here. The Figure can be viewed in the relevant Word document.)

Reference:

Craney, T. A., & Surles, J. G. (2002). Model-Dependent Variance Inflation Factor Cutoff Values. Quality Engineering, 14(3), 391–403. https://doi.org/10.1081/QEN-120001878

O’brien, R.M. A Caution Regarding Rules of Thumb for Variance Inflation Factors. Qual Quant 41, 673–690 (2007). https://doi.org/10.1007/s11135-006-9018-6

Kim, J. H. (2019). Multicollinearity and misleading statistical results. Korean journal of anesthesiology, 72(6), 558-569.

Response to Reviewer #2

Reviewer #2: “The abstract is slightly long and statistically dense. Abstract could be made little simpler..”

Answer: Thank you for your valuable feedback. We have revised the abstract to be more concise and less statistically dense. We believe this version is simpler and more accessible to readers.

Reviewer #2: “This is a cross-sectional study, but at several places the wording sounds like presbyphagia leads to urinary incontinence. Please be careful with language-Say “associated with”, not “predicts”, “affects”, or “should be screened as standard practice”. The conclusions and clinical suggestions should be more cautious and framed as hypothesis-generating, not practice-changing.”

Answer: a) Thank you for your insightful comment regarding the language used in our manuscript. You are absolutely right that in a cross-sectional study, terminology should reflect associations rather than causal relationships. Some of these wording issues arose during the translation process from our native language to English. We have now carefully revised the manuscript accordingly. For example:

Original: "Multiple linear analysis was performed to predict the UI dependent variables using the independent variable of presence/absence of presbyphagia, age and gender."

Revised: "Multiple linear regression analysis was conducted to examine the association between UI and the independent variables: presence/absence of presbyphagia, age, and gender."

b) In response to your comment, we have carefully revised the conclusion and discussion section accordingly. The overall tone of the conclusion now reflects a more cautious and hypothesis-generating perspective, highlighting the need for future longitudinal and interventional studies to confirm our findings.

Reviewer #2: Definition of presbyphagia needs clearer justification

Answer: We have substantially revised the introduction to provide a clearer, more comprehensive definition and justification of presbyphagia. The revised text now:

• Provides a precise, evidence-based definition of presbyphagia (structural, physiological, and innervational alterations)

• Specifies the underlying mechanisms (brain atrophy, neurological deterioration, muscle loss)

• Maintains connection to clinical symptoms with appropriate citations

We believe this revision addresses the reviewer's concern by clearly justifying what presbyphagia is and how it develops.

Reviewer #2: statistical analysis: The logistic regression model used to calculate the odds ratio is not clearly described Confidence intervals for the odds ratio are missing. Clearly state which analysis was used for the odds ratio. Report adjusted OR with 95% CI. Be consistent in how UI is analysed and interpreted

Answer: We thank the reviewers for their valuable comments regarding the odds ratio calculation and presentation. We have carefully addressed these points as follows:

• Removal of unclear odds ratio: The previous version of the manuscript included a statement: *'Participants were classified as having/not having UI according to the ICIQ-SF cut-off value and an odds ratio of 4.61 was calculated.'* We agree with the reviewers that this statement was insufficiently detailed, lacking information about the statistical method used, confidence intervals, and adjustment for confounders. This unclear statement has been removed entirely from the revised manuscript.

• Proper logistic regression analysis: In its place, we have now conducted and reported a binary logistic regression analysis following standard methodological guidelines. In the Statistical Analysis section, we clearly describe:

Variable coding: presbyphagia (0 = non-presbyphagia, 1 = presbyphagia), UI presence (ICIQ-SF ≥6; 0 = no UI, 1 = UI present)

Covariates: age and gender

Exclusion of handgrip strength due to high multicollinearity with gender (VIF > 5, r = -0.96)

• Complete reporting of results: In the Results section, we now present the adjusted odds ratio with its 95% confidence interval: OR = 7.47 (95% CI: 2.24–24.94, p = 0.001). A detailed table (Table 3) following simplified APA format now includes B coefficients, standard errors, Wald statistics, p-values, odds ratios, and confidence intervals for all variables. Model fit statistics (χ² = 21.53, df = 3, p < 0.001; Cox & Snell R² = 0.26; Nagelkerke R² = 0.35; correct classification = 71.2%) are also reported.

• Addressing confidence interval width: We acknowledge that the confidence interval (2.24–24.94) is relatively wide. This limitation is now addressed in the Discussion section, where we note that the wide CI likely reflects the relatively small sample size and the exploratory nature of this study. Importantly, the lower bound remains substantially above 1 (2.24), supporting the robustness of the finding.

Reviewer #2: Figure 1 needs a clearer explanation in the text

Answer: "Thank you for your comment regarding Figure 1. We have revised the manuscript to provide a clearer explanation of the figure in the text.

In the revised Results section, we now state:

*'Based on the ICIQ-SF question "When does urine leak?", 16.44% of participants (n=12) reported no urine leakage. Among the 83.56% of participants who experienced some form of incontinence, the most frequently reported circumstances were: leakage when asleep (20.55%, n=15), leakage when coughing or sneezing (17.81%, n=13), leakage during physical activity (15.07%, n=11), leakage after finishing urination (9.59%, n=7), leakage before reaching the toilet (8.22%, n=6), constant leakage (6.85%, n=5), and leakage for no obvious reason (5.48%, n=4). The distribution of urinary incontinence patterns is illustrated in Figure 1.'*

Additionally, we have expanded the figure caption to: *'Distribution of urinary incontinence patterns among nursing home residents (n=73). Based on ICIQ-SF item: "When does urine leak?" Values represent percentage of the total sample and number of participants.'*

We believe these revisions provide a clearer and more comprehensive explanation of Figure 1 as requested."

Reviewer #2: Tables are very dense and heavy on abbreviations

Answer: We have revised all tables to improve clarity and reduce density. Simplified table footnotes by removing redundant abbreviations and providing only essential information in a clear, concise format. Combined the unstandardized regression coefficient (B) and its 95% confidence interval into a single column (B [95% CI]) to streamline presentation and enhance readability.

Reviewer #2: UI and swallowing function can also be influenced by:BMI, Mobility, Comorbidities, Hydration status, Medication burden.If these variables are unavailable, please explicitly acknowledge residual confounding as a limitation. If some data exist (e.g., BMI), consider including them in an exploratory model.

Answer:

• Handgrip strength (sarcopenia): In response to reviewer comments regarding potential confounding by generalized muscle weakness and sarcopenia, we obtained handgrip strength data from institutional records—as these measurements are routinely collected every three months in our nursing home—and attempted to include this variable in our regression models as a proxy for sarcopenia. However, during diagnostic testing for multicollinearity, we encountered a significant statistical issue. The correlation matrix revealed an extremely high negative correlation between handgrip strength and gender (r = -.959), indicating that these two variables shared almost identical variance. Consequently, the Variance Inflation Factor (VIF) values for gender and handgrip strength were 12.55 and 12.54, respectively—far exceeding the recommended threshold of 5 [Craney and Surles 2002; O’brien 2007]. As noted by Craney and Surles (2002), VIF values above 5 indicate problematic multicollinearity that can distort regression coefficients, inflate standard errors, and compromise the reliability and validity of statistical models. Following established methodological recommendations [Kim 2009], researchers facing high multicollinearity have several options: remove one of the correlated variables, use ridge regression, or combine variables into a single index. Given that gender is a fundamental demographic variable with strong theoretical justification for inclusion, and given the exceptionally high correlation between handgrip strength and gender, we made the considered decision to exclude handgrip strength from the final models. This approach ensures stable, interpretable, and reliable regression results. We have now explicitly acknowledged in the Limitations section that, despite our attempt to include handgrip strength, sarcopenia could not be fully controlled for as a potential confounder due to these statistical constraints.

Reference: Craney, T. A., & Surles, J. G. (2002). Model-Dependent Variance Inflation Factor Cutoff Values. Quality Engineering, 14(3), 391–403. https://doi.org/10.1081/QEN-120001878

O’brien, R.M. A Caution Regarding Rules of Thumb for Variance Inflation Factors. Qual Quant 41, 673–690 (2007). https://doi.org/10.1007/s11135-006-9018-6

Kim, J. H. (2019). Multicollinearity and misleading statistical results. Korean journal of anesthesiology, 72(6), 558-569.

Figure 1. Correlations (handgrip vs gender) (The Figure file format is not supported, so it could not be added here. The Figure can be viewed in the relevant Word document.)

Figure 2. VIF value (The Figure file format is not supported, so it could not be added here. The Figure can be viewed in the relevant Word document.)

• BMI: We chose not to include BMI in the final models for several methodological reasons. First, to avoid increasing Type II error risk: Adding BMI as an additional covariate would increase the number of statistical tests, thereby lowering the significance threshold when applying multiple comparison corrections (e.g., Bonferroni). This could obscure truly meaningful associations. Second, with a sample size of 73, including too many covariates risks overfitting and unstable estimates (Pergener, 1998). Following principles of parsimonious modeling, we prioritized variables with stronger theoretical justification: presbyphagia, age, and gender. Third, preliminary bivariate analyses showed no significant association between BMI and our primary outcomes, further supporting this decision. We believe this balanced approach maintains statistical power while appropriately addressing confounding.

Referance: “Perneger TV. What's wrong with Bonferroni adjustments. BMJ. 1998 Apr 18;316(7139):1236-8. doi: 10.1136/bmj.316.7139.1236. PMID: 9553006; PMCID: PMC1112991.

• Mobility, comorbidities, and medication burden: These potential confounders were addressed through rigorous a priori exclusion criteria, which we believe strengthens our methodological approach.

Mobility: Individuals with a history of falls or requiring ambulation assistance were excluded (criterion #2), ensuring a functionally independent sample with preserved mobility.

Comorbidities: We excluded individuals with neurological/psychiatric diseases (criterion #3), conditions affecting swallowing/UI mechanisms (criterion #4), and acute systemic illnesses (criterion #6), thereby reducing confounding from major comorbidities.

Medication burden: Regular use of medications affecting urinary function or cognition (e.g., diuretics, anticholinergics) was an exclusion criterion (criterion #5), eliminating pharmacological confounding from these classes.

• Hydration : We acknowledge that hydration status was not directly measured in this study, and we have now added this as an explicit limitation in the revised manuscript. Future studies should incorporate objective measures of hydration status to further clarify its potential role in the presbyphagia-UI relationship.

Response to Reviewer #3

Reviewer #3: This is an original study that raises an interesting but conceptually challenging question. The primary aim is to demonstrate an association between presbyphagia and urinary incontinence. While such an association is statistically demonstrated, causation is not established, and the clinical or practical implications of this finding remain unclear. The results are statistically significant but conceptually weak in terms of mechanistic or clinical interpretation.

Answer: We thank the reviewer for th

---

## [Decision Letter · Decision Letter 1]

8 Apr 2026

PONE-D-25-62171R1The Association of Presbyphagia with Urinary Incontinence and Incontinence-Related Quality of Life in Nursing Home Residents: A Cross-Sectional StudyPLOS One

Dear Dr.  Yıldız,

Thank you for submitting your manuscript to PLOS ONE. After careful consideration, we feel that it has merit but does not fully meet PLOS ONE’s publication criteria as it currently stands. Therefore, we invite you to submit a revised version of the manuscript that addresses the points raised during the review process.

We look forward to receiving your revised manuscript.

Kind regards,

Stanisław Jacek Wroński, M.D., Ph.D, FEBU

Academic Editor

PLOS One

Journal Requirements:

Reviewer's Responses to Questions

**Comments to the Author**

1. If the authors have adequately addressed your comments raised in a previous round of review and you feel that this manuscript is now acceptable for publication, you may indicate that here to bypass the “Comments to the Author” section, enter your conflict of interest statement in the “Confidential to Editor” section, and submit your "Accept" recommendation.

Reviewer #1: All comments have been addressed

Reviewer #2: All comments have been addressed

Reviewer #3: All comments have been addressed

2. Is the manuscript technically sound, and do the data support the conclusions?

Reviewer #1: Yes

Reviewer #2: Yes

Reviewer #3: Yes

3. Has the statistical analysis been performed appropriately and rigorously? 

Reviewer #1: Yes

Reviewer #2: Yes

Reviewer #3: Yes

4. Have the authors made all data underlying the findings in their manuscript fully available?

Reviewer #1: Yes

Reviewer #2: Yes

Reviewer #3: Yes

5. Is the manuscript presented in an intelligible fashion and written in standard English?

Reviewer #1: Yes

Reviewer #2: Yes

Reviewer #3: Yes

6. Review Comments to the Author

Reviewer #1: First, I want to congratulate the authors for their dedication in revising this manuscript. The effort made to effectively improve the work is evident.

The methodological and statistical adjustments made were pertinent and fully addressed my previous assessment.

There are some minor details throughout the text that appear to be typos and deserve attention (they are indicated in the PDF).

I suggest that when citing only one author in a paragraph, the identification should be unique to avoid tiring the reader; therefore, I suggest a careful review of the writing.

I hope to read this published work soon.

Reviewer #2: The authors have made a clear effort to address the concerns raised in the previous round of review, and the manuscript has improved in terms of clarity, statistical reporting, and the use of appropriate associative language. The research question is relevant, and the use of validated instruments strengthens the methodological approach.

Overall, the study is methodologically sound, and the analyses are appropriate for the cross-sectional design. The results are clearly presented, and the conclusions are generally supported by the data at the level of association.

However, a few points would benefit from further clarification and refinement:

* Although some confounders have been considered, there remains the possibility of residual confounding from factors such as sarcopenia, frailty, or overall functional decline. While these are now acknowledged, their potential influence on the observed associations should be more consistently emphasized in the discussion and conclusion.

*Both the exposure and outcome measures rely on self-reported instruments, which introduces the possibility of shared method bias or general symptom reporting tendencies. This limitation should be highlighted more explicitly.

*Urinary incontinence is treated as a single entity, whereas different subtypes may have distinct underlying mechanisms. This should be more clearly acknowledged as a limitation when interpreting the findings.

*The manuscript would benefit from minor language editing to improve clarity and readability, as there are occasional grammatical errors and instances of awkward phrasing.

Reviewer #3: I would like to thank the authors for the huge effort made to improve this manuscript. The only suggestion is to define in the first line of the manuscript Urinary incontinence. You are using the abbreviation strait away. I know it's used in th abstract, but the manuscript and abstract a different entities.

7. PLOS authors have the option to publish the peer review history of their article (what does this mean?). If published, this will include your full peer review and any attached files.

Reviewer #1: **Yes:**Denise Pinheiro Marques Alves dos Santos

Reviewer #2: **Yes:**Amisha S Amin

Reviewer #3: No

---

## [Author Response · Author response to Decision Letter 2]

22 Apr 2026

RESPONCE TO REVİEWERS 1

Reviewer 1: “First, I want to congratulate the authors for their dedication in revising this manuscript. The effort made to effectively improve the work is evident.

The methodological and statistical adjustments made were pertinent and fully addressed my previous assessment…, …. I hope to read this published work soon...”

Answer: We sincerely thank the reviewer for their encouraging and generous comments. We are delighted to hear that the methodological and statistical adjustments were considered pertinent and that our revisions have fully addressed the previous concerns. It is a great pleasure to know that the reviewer hopes to read our published work soon. We greatly appreciate their time and constructive input throughout the review process.

Reviewer 1: “…There are some minor details throughout the text that appear to be typos and deserve attention (they are indicated in the PDF)...” in PDF: “Typo? The methodology specifies a maximum age of 95 years.”

Answer: We thank the reviewer for noting this point. The inclusion criteria specified an age range of 65–95 years; however, the oldest participant in our sample was 92 years. This is not a typo but simply reflects the actual age distribution of the residents who met all inclusion criteria during the study period. We have confirmed that the text is accurate and no correction is needed. We appreciate the reviewer’s careful attention to detail.

Reviewer 1: “…I suggest that when citing only one author in a paragraph, the identification should be unique to avoid tiring the reader; therefore, I suggest a careful review of the writing…”

Answer: We thank the reviewer for this valuable suggestion. Accordingly, we have revised the two paragraphs concerning De Jesus Fidyk et al. and Matsumoto et al. In each paragraph, the reference is now cited only once at the beginning, with subsequent mentions replaced by pronouns such as "they" or "their study" to avoid unnecessary repetition. We appreciate the reviewer’s careful reading.

RESPONCE TO REVİEWERS 2

Reviewer 2: “The authors have made a clear effort to address the concerns raised in the previous round of review, and the manuscript has improved in terms of clarity, statistical reporting, and the use of appropriate associative language. The research question is relevant, and the use of validated instruments strengthens the methodological approach.

Overall, the study is methodologically sound, and the analyses are appropriate for the cross-sectional design. The results are clearly presented, and the conclusions are generally supported by the data at the level of association…”

Answer: We thank the reviewer for their positive and constructive feedback. We are glad that the revisions have met their expectations.

Reviewer 2: “However, a few points would benefit from further clarification and refinement:

* Although some confounders have been considered, there remains the possibility of residual confounding from factors such as sarcopenia, frailty, or overall functional decline. While these are now acknowledged, their potential influence on the observed associations should be more consistently emphasized in the discussion and conclusion…”

Answer: We have now more consistently highlighted the potential influence of residual confounding (e.g., sarcopenia, frailty) in both the Discussion and Conclusion sections. However, we deliberately avoided a detailed comparison between these factors and swallowing dysfunction in the Discussion, as it would have disrupted the logical flow of the manuscript and made this section excessively long.

Reviewer 2: “*Both the exposure and outcome measures rely on self-reported instruments, which introduces the possibility of shared method bias or general symptom reporting tendencies. This limitation should be highlighted more explicitly.”

Answer: We have now explicitly highlighted the possibility of shared method bias and general symptom reporting tendencies in the Limitations section (third point), as suggested. The revised text clarifies that the use of symptom‑based questionnaires for both exposure and outcomes may have inflated the observed associations.

Reviewer 2: *Urinary incontinence is treated as a single entity, whereas different subtypes may have distinct underlying mechanisms. This should be more clearly acknowledged as a limitation when interpreting the findings.

Answer: We have now elaborated on this limitation in more detail in the Limitations section, explicitly noting that stress and urgency UI may involve distinct pathophysiological mechanisms, which limits the interpretation of our findings. Thank you for the suggestion.

Reviewer 2: *The manuscript would benefit from minor language editing to improve clarity and readability, as there are occasional grammatical errors and instances of awkward phrasing.

Answer: The suggestion has been taken into account. The article has been reviewed again and the grammatical errors have been corrected.

RESPONCE TO REVİEWERS 3

Reviewer 3: “We thank the reviewer for this suggestion. We have now defined "urinary incontinence (UI)" at the very beginning of the Introduction section, before using the abbreviation. The first sentence of the manuscript has been revised accordingly.”

Answer: We thank the reviewer for this suggestion. We have now defined "urinary incontinence (UI)" at the very beginning of the Introduction section, before using the abbreviation. The first sentence of the manuscript has been revised accordingly.

---

## [Decision Letter · Decision Letter 2]

21 May 2026

The Association of Presbyphagia with Urinary Incontinence and Incontinence-Related Quality of Life in Nursing Home Residents: A Cross-Sectional Study

PONE-D-25-62171R2

Dear Dr. Ziya Yildiz

we’re pleased to inform you that your manuscript has been judged scientifically suitable for publication and will be formally accepted for publication once it meets all outstanding technical requirements.

Kind regards,

Stanisław Jacek Wroński, M.D., Ph.D, FEBU

Academic Editor

PLOS One

Reviewer's Responses to Questions

**Comments to the Author**

1. If the authors have adequately addressed your comments raised in a previous round of review and you feel that this manuscript is now acceptable for publication, you may indicate that here to bypass the “Comments to the Author” section, enter your conflict of interest statement in the “Confidential to Editor” section, and submit your "Accept" recommendation.

Reviewer #1: All comments have been addressed

Reviewer #2: All comments have been addressed

Reviewer #3: All comments have been addressed

2. Is the manuscript technically sound, and do the data support the conclusions?

Reviewer #1: Yes

Reviewer #2: Yes

Reviewer #3: Yes

3. Has the statistical analysis been performed appropriately and rigorously? 

Reviewer #1: Yes

Reviewer #2: Yes

Reviewer #3: Yes

4. Have the authors made all data underlying the findings in their manuscript fully available?

Reviewer #1: Yes

Reviewer #2: Yes

Reviewer #3: Yes

5. Is the manuscript presented in an intelligible fashion and written in standard English?

Reviewer #1: Yes

Reviewer #2: Yes

Reviewer #3: Yes

6. Review Comments to the Author

Reviewer #1: I would like to acknowledge the effort and the quality of the revisions undertaken. The level of rigor applied throughout the process substantially improved the clarity, coherence, and methodological robustness of the manuscript, rendering it suitable for publication. The final version reflects a well‑structured and technically sound contribution.

Reviewer #2: The manuscript has improved substantially compared to the previous version, and the authors have addressed the major methodological, statistical, and interpretative concerns raised during earlier rounds of review. The discussion and conclusions are now more balanced, and the limitations have been appropriately acknowledged.

The study is scientifically sound and suitable for publication. However, before final acceptance, a final round of careful language and editorial proofreading is recommended to correct minor grammatical issues, typographical errors, spacing inconsistencies, and formatting problems throughout the manuscript. Additionally, please review the statement in the conclusion regarding handgrip strength adjustment, as handgrip strength was excluded from the final regression model due to multicollinearity.

Overall, I believe the manuscript is acceptable after minor editorial revisions.

Reviewer #3: In my opinion the article is mature enough to be published. All issues have been addressed and it is logical and readable now.

7. PLOS authors have the option to publish the peer review history of their article (what does this mean?). If published, this will include your full peer review and any attached files.

Reviewer #1: **Yes:**Denise Pinheiro Marques Alves dos Santos

Reviewer #2: **Yes:**Amisha S Amin

Reviewer #3: No

---

## [Editor Report · Acceptance letter]

PONE-D-25-62171R2

PLOS One

Dear Dr. Yıldız,

I'm pleased to inform you that your manuscript has been deemed suitable for publication in PLOS One. Congratulations! Your manuscript is now being handed over to our production team.

Kind regards,

on behalf of

Dr. Stanisław Jacek Wroński

Academic Editor

PLOS One